# An Eco-Friendly Quaternary Ammonium Salt as a Corrosion Inhibitor for Carbon Steel in 5 M HCl Solution: Theoretical and Experimental Investigation

**DOI:** 10.3390/molecules27196414

**Published:** 2022-09-28

**Authors:** Rem Jalab, Mohammed A. Saad, Mostafa H. Sliem, Aboubakr M. Abdullah, Ibnelwaleed A. Hussein

**Affiliations:** 1Gas Processing Center, College of Engineering, Qatar University, Doha P.O. Box 2713, Qatar; 2Chemical Engineering Department, College of Engineering, Qatar University, Doha P.O. Box 2713, Qatar; 3Center for Advanced Materials, Qatar University, Doha P.O. Box 2713, Qatar;p

**Keywords:** eco-friendly surfactant, corrosion inhibitor, quantum calculations

## Abstract

The corrosion of industrial material is a costly problem associated with global economic losses reaching trillions of US dollars in the repair of failures. Injecting corrosion inhibitors is the most practically promising method for decelerating corrosion reactions and protecting surfaces. Recent investigations have focused on surfactants as corrosion inhibitors due to their amphiphilic nature, low cost, and simple chemical preparation procedures. This study aims to investigate the performance of an environment-friendly Quaternium-22 (Q-22) surfactant which is widely used in cosmetics for C-steel corrosion inhibition in a 5 M HCl medium. Weight loss experiments were performed at different concentrations and immersion times, presenting a maximum efficiency at 2.22 mmol·L^−1^. The influence of Q-22 on the corrosion behavior of C-steel was elucidated using non-destructive electrochemical measurements. The overall results revealed that adding varied concentrations of Q-22 significantly decreases the corrosion rate of C-steel. The results revealed the physisorption nature of Q-22 onto the C-steel surface, with adsorption following the Freundlich isotherm (∆$H_{\mathrm{ads}}$= −16.40 kJ·mol^−^^1^). The relative inhibition performance of Q-22 was also evaluated by SEM and AFM analyses. Lastly, quantum chemical calculations based on density functional theory (DFT) demonstrated that Q-22 has promising molecular features concerning the anticorrosive mechanism.

## 1. Introduction

Corrosion in oil and gas wells is a persistent challenge arising from the existence of many corrosive impurities, including saline water, hydrogen sulfide (H_2_S), and carbon dioxide (CO_2_), commonly transported with natural gas or crude oil. Additionally, the application of wells’ acidizing treatments to dissolve components clogging the flow of oil and gas involves pumping corrosive acids. For instance, regularly injected hydrochloric (HCl), hydrofluoric (HF), sulfuric (H_2_SO_4_), sulfurous (H_2_SO_3_), acetic (CH_3_COOH), and formic (HCOOH) acids stimulate rapid corrosion development [1,2]. Indeed, corrosion is costly, as the production processes becomes complicated until all failures are repaired. Corrosion damaging consequences include chemical leakage, the breakdown pipelines and machines, and metallic equipment failures [3]. Currently, the global costs of corrosion failures are estimated to reach approximately 2.5 trillion USD annually, comparable to 3.4% of the world’s gross domestic product (GDP) [1,4,5,6]. The replacement, repair, painting of corroded equipment, and use of alloys and high resistance materials are considered direct economic losses. On the other hand, the shutdown of operating plants, pauses in production, environmental pollution, and losses in equipment efficiency are indirect losses that are difficult to estimate quantitatively [1].

The corrosion rate can be predominantly influenced by the nature of the metal and the corroding environment conditions, such as the pH, temperature, and air humidity [7]. Carbon steel (C-steel) pipelines are extensively used in many industrial applications, specifically for transporting oil and gas in production facilities due to their cost-effectiveness and promising performance [8]. However, C-steel is characterized by possessing a high corrosion susceptibility [9]. Yet, microalloying with carbides or nitrides can improve its corrosion resistance [10]. Referring to NACE MR0175 Standards and ISO 13680, the use of corrosion-resistant alloys is superior in terms of prevention in the long term for different corrosion types in the downhole of oil and gas wells compared to C-steel [11,12].

Moreover, a practical corrosion mitigation strategy is the injection of corrosion inhibitors (CIs) to protect the targeted large surfaces. CIs act through inhibitory mechanisms, classified into film-forming, scavenging, and neutralizing mechanisms, by lowering the H^+^ concentration, relying on their chemical composition [13]. Corrosion inhibitors are either organic types of inhibitors that get adsorbed onto the metal surface and decelerate corrosion or inorganic ones that react with anodic and cathodic elements and are deposited, forming a barrier coating [4,14]. It is worth mentioning that competent CIs with strong inhibition efficiency have heteroatoms (N, O, and P), aromatic functional groups, and π electrons that serve active sites, facilitating their adsorption onto the metal surface [15,16]. Conducting polymers (CPs) are an example of efficient corrosion inhibitors, containing repeating units, which serve many active sites and cover a large surface area [17]. Thus, CPs coatings have been widely used since they forms protective films [18].

Current approaches for corrosion inhibitors are directed towards the use of polymers possessing a large surface area, with several binding locations and polar functional groups such as hydroxyl, carboxylic, amine, and aromatic groups [19]. Furthermore, surfactants have been commonly selected as corrosion inhibitors to prevent metallic corrosion, benefitting from their amphiphilic nature, which allows for adsorption at interfaces [20]. Their promising inhibition efficiency, low toxicity, low cost, and simple synthesis and production are all features that laid the foundations for using surfactants in corrosion inhibition applications [21]. Surfactants are broadly used in various applications, such as in cleaning products (detergents, soaps), shampoos, coating additives, and paints in industries. There are several types of surfactants: anionic, cationic, nonionic, and amphoteric. The unique characteristics of surfactants and the presence of nitrogen atoms with free electron pairs as part of the functional group facilitate adsorption and formation of bonds with the metal surface.

Recently, the effectiveness of surfactants as corrosion inhibitors has been widely investigated in different mediums and for the protection of several metals, contributing to approximately 24% of organic-based corrosion inhibitor studies [22,23,24]. As new generation surfactants of quaternized salts, consisting of two surfactant monomers linked together through a spacer group, Gemini surfactants have been extensively examined in corrosion studies [25,26]. For instance, novel cationic Gemini surfactants synthesized for corrosion inhibition in the context of C-steel pipelines in 1 M HCl medium by Hegazy et al. [27] exhibited a strong inhibition efficiency, reaching 93.7%. Additionally, three cationic Gemini surfactants with varying spacer lengths named G-12, G-6, and G-2 illustrated the effect of a lengthy spacer chain on improving the surface coverage [21]. Moreover, other cationic Gemini surfactant compounds showed corrosion inhibition efficiencies between 76–81% at 300 ppm concentration in oil well formation water with existing sulfide ions [28]. Nonionic Gemini surfactants based on adipic acid with a varying number of propylene oxide units demonstrated a 99.4% corrosion inhibition efficiency for C-steel in 1 M HCl [29]. Based on weight loss data, the adsorption of these surfactants was best described by the Langmuir isotherm model. Other novel green surfactants synthesized from erucic acid were tested for mild steel corrosion inhibition in 15% HCl solution [30]. They exhibited a 98.9% inhibition efficiency at a temperature of 90 °C. Cationic surfactants synthesized with a Schiff base group reached 95% protection efficiency for C-steel in 3.5% NaCl + 0.5 M HCl medium in the range of 30–60 °C [31]. Additionally, anionic surfactants, namely Diisononyl phthalate, Noleyl-1, 3-propane –diamine, and Sodium lauryl sulfate, were also investigated for the corrosion of C-steel in 1 M HCl solution and demonstrated efficiencies of 85.6, 84, and 39.2% at 300 mg·L^−1^ concentration [32]. Four different eco-friendly nonionic surfactants named Triton X-100, Tween 20, Tween 80, and Brij 35 were examined for C-steel in 1 M HCl and compared with the cationic surfactant cetyltrimethylammonium bromide (CTAB) [33]. The study revealed that these inhibitors have comparable performances and recorded efficiencies in the range of 91–92% at 30 °C compared to 97% for CTAB.

The present study aimed to investigate Quaternium-22 (Q-22) cationic surfactant as an environment-friendly corrosion inhibitor for C-steel pipelines in oil and gas fields. Q-22 is a quaternary ammonium compound known to be extensively used in cosmetics and personal care products and has not been tested before in any corrosion study. The corrosion experiments were carried out at ambient and high temperatures in a 5 M HCl medium, representing the harsh conditions of a well’s acidizing treatment for production improvement. The Q-22 was investigated by gravimetric analysis, electrochemical impedance spectroscopy (EIS), and potentiodynamic polarization (PDP) techniques. This work also studied the adsorption isotherms and corrosion kinetics and determined all related parameters. Lastly, molecular simulation using density functional theory (DFT) was employed to demonstrate the performance of Q-22 and to determine the relationship between its anticorrosive mechanism and its chemical structure.

## 2. Materials and Methods

### 2.1. Materials

Quaternium-22 (Q-22) surfactant was supplied by Shanghai Dejun Technology Co., Ltd., Shanghai, China. Q-22 is an eco-friendly surfactant used as a cosmetic and/or antistatic agent, with a chemical formula of C_13_H_29_N_2_ClO_7_ and a molecular structure shown in Figure 1. Different concentrations of the inhibitor (200, 400, 600, 800 mg·L^−^^1^) or (0.55, 1.11, 1.66. 2.22 mmol·L^−^^1^) were prepared in deionized water. The corrosive solution was 5 M HCl, prepared from the dilution of analytical grade 37% HCl using deionized water. The C-steel coupons were supplied by Qatar Steel Co., Ltd., Doha, Qatar, and were cut from AISI 1020 alloy sheets. The chemical composition of the coupons was iron and 0.2% carbon with up to 0.7% manganese, 0.65% silicon, and 0.65% copper in wt.%. The coupons were cut and ground using silicon carbide (SiC) abrasive papers and were abraded to a 4000 grit finish. Following that, the coupons were washed with deionized water and dried in the oven at a temperature of 150 °C.

### 2.2. Weight Loss Measurements

C-steel coupons with dimensions of 1.5 × 1.5 × 0.1 cm were used for the weight loss experiments. The coupons were immersed in 200 mL solutions of 5 M HCl with and without different concentrations of the Q-22 inhibitor. The tests were performed at room temperature and for three different time durations of 30, 120, and 240 min. The weight of the dry coupons was reported before and after immersion. The inhibition efficiency was determined according to the following (Equation (1)):(1)$$\mathrm{IE}\;\%=Ɵ\times 100=\frac{W_{0}-W_{1}}{W_{0}}\times 100\%$$
where Ɵ is the surface coverage and $W_{0}$ and $W_{1}$ are the weight loss measurements with and without the inhibitor, respectively.

The corrosion rate can also be found according to (Equation (2)):(2)$$\mathrm{Corrosion}\;\mathrm{rate}\;\left(\mathrm{mpy}\right)=\frac{534\;W}{\;\rho \;A\;t}$$
where W is the mass loss in mg, ρ is the C-steel density in g·cm^−^^1^, A is the surface area of the coupon in cm^2^, and t is the time in an hour.

The tests were repeated three times for checking the reproducibility of the data, and the average measurements were reported.

### 2.3. Electrochemical Measurements

The electrochemical measurements were obtained by GAMRY 3000 potentiostat/galvanostat/ZRA (Warminster, PA, USA) using a double-jacketed glass cell. The experiments were performed at different temperatures (20, 30, 50, and 70 °C), and the temperatures were controlled using a Julabo thermostat (GmbH, Seelbach, Germany). In the three-electrode cell, a graphite rod was used as a counter electrode and a standard calomel electrode (SCE) as a reference electrode. C-steel sheets acting as working electrodes have a 0.5 cm^2^ cross-sectional area exposed to the electrolyte solution. The corrosion of C-steel specimens was investigated against 5 M HCl solutions prepared with and without various concentrations of Q-22 inhibitor (200, 400, 600, and 800 mg·L^−1^ or 0.55, 1.11, 1.66. 2.22 mmol·L^−^^1^). Initially, the C-steel specimens were put under open-circuit conditions for around 20 min. Electrochemical impedance spectroscopy (EIS) tests were carried out under a frequency ranging from 0.1 Hz to 100 kHz and an AC amplitude of 10 mV and 10 points/decades. The Potentiodynamic polarization curves of the C-steel were attained from −250 to +250 mV against the open circuit potential at a scan rate of 0.3 mV·s^−1^. All sets of experiments were conducted three times to check how successful the system was in providing reproducible data. Therefore, the data were reproducible with less than 5% errors, and the average measurements were reported.

### 2.4. Surface Characterization

The surface topography of carbon steel before and after the corrosion was studied using a field emission scanning electron microscope (FE-SEM, FE-SEM-Nova Nano-450, The Netherlands). Additionally, the Asylum Research MFP-3D atomic force microscope (AFM, Santa Barbara, CA, USA) was used to measure the surface roughness and surface topography in nanoscale with a non-contact mode.

### 2.5. Eco-Toxicity Assessment

The ADMETSAR 2 program was employed for the evaluation of the eco-toxic properties of Q-22 inhibitor molecules [34,35]. This program assesses the absorption, distribution, metabolism, excretion, and toxicity (ADMET) parameters. The web tool is based on a machine-learning model formulated from more than 210,000 experimental outcomes for 100,000 chemical compounds.

### 2.6. Quantum Chemical Studies

The quantum chemical calculations were based on the Density Functional Theory (DFT) method. Changes in the electronic structure responsible for the inhibition properties of the molecule could be best described by the DFT. The required input files for structure optimization and frequency calculation through the DFT simulations were prepared using Gaussian 09 software [36]. The ground-state DFT and the methods of Becke’s three parameters, Lee, Yang, and Parr (DFT-B3LYP), with a 6-311+g(d,p) basis set, were used for the calculations. These methods are recognized by producing an accurate determination of reactivity properties [37]. The optimization of the Q-22 molecule was achieved following a gradient minimization technique. The vibration analysis was carried without imaginary frequencies to guarantee a minimal energy state with respect to the optimized molecules. All the essential quantum parameters describing the molecular interactions were calculated, and the density graphical isosurfaces were visualized.

The energy of the highest occupied molecular orbital ($E_{\mathrm{HOMO}}$) and the lowest unoccupied molecular orbital ($E_{\mathrm{LUMO}}$) acquired from the optimized output files are key indicators of electronic parameters. Quantum parameters were determined by relying on Koopman’s theorem, stating that the ionization energy $\left(I\right)$ correlates to $\left(I=-E_{\mathrm{HOMO}}\right)$ and electron affinity to $\left(A=-E_{\mathrm{LUMO}}\right)$ [38,39]. Other reactivity parameters, including chemical hardness $\left(\eta \right)$, electronegativity $\left(\chi \right)$, potential $\left(\mu \right)$, and electronegativity index (ω) are obtained relying on the values of I and A (Equations (3)–(6)) [40,41]. The total negative charge (TNC) parameter, demonstrating the available adsorption sites on the molecule, was estimated from Mulliken charges [42].
(3)$$\eta =\frac{I-A}{2}$$
(4)$$\chi =\frac{I+A}{2}$$
(5)μ=−χ
(6)$$\omega =\frac{\mu^{2}}{2\eta }$$

## 3. Results and Discussion

### 3.1. Weight Loss Measurements

Figure 2 illustrates the weight loss transients of C-steel specimens immersed in 5 M HCl solution with and without different concentrations of Q-22 inhibitor at 25 °C. The impact of inhibitor concentration was studied by adding 200, 400, 600, and 800 mg·L^−1^ (0.55, 1.11, 1.66, and 2.22 mmol·L^−1^) of Q-22 inhibitor. The results in Figure 2 demonstrate the mass loss decline of the C-steel with the increase in inhibitor concentration. The higher the Q-22 concentration, the better the adsorption potentials of Q-22 on the C-steel surface, hence the higher inhibition efficiency (IE%) (Table 1). The adsorption of Q-22 onto the C-steel surface was facilitated by the interaction of iron atoms with the lone electron pairs of oxygen atoms existing through the chain of Q-22. Increased IE% values are reported in Table 1 at higher concentrations, indicating the higher surface coverage and adsorption of inhibitor molecules onto the surface [43].

Furthermore, the effect of C-steel immersion time on the corrosion was examined in the presence and absence of various concentrations over 0.5, 2, and 4 h. The results of reduced efficiency confirm the importance of a long immersion time in inducing an entanglement between the inhibitor molecules, thus exposing the active sites of the surface to corrosion [44,45]. At the highest concentration of Q-22 inhibitor (2.22 mmol·L^−1^), a maximum efficiency of 56% was achieved upon immersion of C-steel for 0.5 h. However, after 4 h of immersion in the same concentration, the efficiency of the inhibitor had declined by approximately 58%, reaching 23%.

### 3.2. Electrochemical Measurements

#### 3.2.1. Electrochemical Impedance Spectroscopy (EIS)

EIS measurements are essential for understanding corrosion mechanisms and acquiring information about electrochemical reaction kinetics [46]. In addition, EIS supports obtaining information regarding the metal surface-solution interface and the effect of the inhibitor on the electric double layer. An electric double layer is formed due to the species aggregation on the metal-solution interface during corrosion. This electric double layer affects the charge transfer between anodic and cathodic sites, affecting corrosion mechanisms. The EIS Nyquist plots (measured and fitted) of C-steel in 5 M HCl with different concentrations of Q-22 inhibitor at temperatures of 20, 30, 50, and 70 °C are shown in Figure 3. The diameter of the semicircles of the Nyquist plots increases at higher concentrations of Q-22 inhibitor. This indicates a stimulated inhibition process upon the addition of more inhibitor molecules [47]. The results at high concentrations of Q-22 demonstrate that the inhibition process becomes effective at such concentrations, possibly due to the formation of a denser protective layer thanks to the Q-22 molecules on the C-steel electrode surface. The Nyquist fitted curves show a figure for a defective heterogeneous surface in aqueous media with a corrosion product and a metal surface response [47]. The semicircles become depressed at higher temperatures [48] and deviate from perfect circular shapes, especially at higher temperatures [49]. The deviation from a circular shape is due to the frequency dispersion of impedance attributed to the roughness arising from the inhomogeneity of the electrode surface or the adsorption of the inhibitor [50,51].

For the analysis of EIS data, a two-time constant equivalent electric circuit (EC) was used, as shown in Figure 4. The EC contained a solution resistance $\left(R_{s}\right)$, the resistance of pores to the corrosion products $\left(R_{1}\right)$, the charge transfer resistance as an inner layer $\left(R_{\mathrm{ct}}\right)$, and constant phase elements from the capacitance of the outer and inner layer (CPE_1_ and CPE_2_), respectively, for fitting the non-ideal double-layer capacitor.

The CPE was defined in impedance according to the following expression:(7)$$Z_{\mathrm{CPE}}=\left(Y_{0}^{-1}{\left(j\omega \right)}^{-n}\right)$$
where $Z_{\mathrm{CPE}}$ is CPE impedance in Ω·cm^−2^, $Y_{0}$ is CPE constant in µs^n^·Ω^−1^·cm^−2^, j = (−1)^1/2^ and ω is angular frequency in rad·s^−^^1^, and n is the measure to surface inhomogeneity, ranging from 0 to 1. An ideal capacitor or ideal resistor is the representative case for CPE when n = 1 or n = 0, respectively. The capacitance of the double layer $\left(C_{\mathrm{dl}}\right)$ was obtained from the following expression:(8)$$C_{\mathrm{dl}}=\frac{{\left(Y_{0}R_{\mathrm{ct}}\right)}^{1/n}}{R_{\mathrm{ct}}}$$

The charge transfer resistance ($R_{\mathrm{ct}}$) is expected to increase more in the solution containing the inhibitor compared to the blank solution. Then, IE% is found from the surface coverage (Ɵ) written in terms of $R_{\mathrm{ct}}$ as:(9)$$\mathrm{IE}\%=Ɵ\times 100=\frac{R_{\mathrm{ct}1}-R_{\mathrm{ct}2}}{R_{\mathrm{ct}1}}\times 100\%$$
where $R_{\mathrm{ct}1}$ and $R_{\mathrm{ct}2}$ are the charge transfer resistance in the presence and absence of the inhibitor, respectively.

All the parameters obtained from the data analysis of EIS Nyquist plots are reported in Table 2. The obtained results assert an increase in the charge transfer resistance ($R_{\mathrm{ct}}$) and a decline in the double-layer capacitance upon raising the Q-22 concentration. Additionally, the increase in the temperature had a pronounced effect on increasing the values of $C_{\mathrm{dl}}$ [45]. These trends were attributed to the rise in the C-steel surface coverage and the enhanced adsorption of inhibitor molecules. It is recognized that the optimum concentration of Q-22, which results in achieving the optimum inhibition efficiency, occurs at the highest $R_{\mathrm{ct}}$ and the lowest $C_{\mathrm{dl}}$ values [52]. Looking at the results, Q-22 at 2.22 mmol·L^−^^1^, as the highest studied concentration, provides the highest $R_{\mathrm{ct}}$, at 19.34 Ω·cm^2^, and lowest $C_{\mathrm{dl}}$, at 284 µF, at 20 °C. Additionally, the highest achieved efficiency at the optimum conditions was 45%. It is obvious that the decrease in $C_{\mathrm{dl}}$ values indicates an increase in the area or the thickness of the electrical double layer. This can be attributed to the increase in the surface roughness and the $R_{\mathrm{ct}}$ values in the presence of the Q-22 corrosion inhibitor. The increase in the charge transfer resistance values may be attributed to either (i) the formed passive film, which is promoted by the presence of the inhibitor molecules that block the active sites on the steel surface, or (ii) the increase in the adsorbed layer thickness/area of the inhibitor, which acts as a physical barrier.

#### 3.2.2. Potentiodynamic Polarization Measurements (PDP)

Figure 5 demonstrates, at a scan rate of 0.3 mV·s^−1^, the potentiodynamic polarization curves of the C-steel specimens in 5 M HCl solution in the presence and absence of different Q-22 concentrations at 20, 30, 50, and 70 °C. The Tafel extrapolation method was used to obtain the electrochemical parameters, including the corrosion current density ($i_{\mathrm{corr}}$) and free potential ($E_{\mathrm{corr}}$), the polarization resistance ($R_{p}$), the corrosion rate (CR), and the anodic ($\beta_{a}$) and cathodic ($\beta_{c}$) Tafel slopes, as shown in Table 3.

The polarization resistance ($R_{p}$) was determined from Stern–Geary equation as follows:(10)$$R_{p}=\frac{\beta_{a}\beta_{c}}{2.303{\;i}_{\mathrm{corr}}\left(\beta_{a}+\beta_{c}\right)}$$

The IE% was determined from the surface coverage (Ɵ), written in terms of $i_{\mathrm{corr}}$ as:(11)$$\mathrm{IE}\%=Ɵ\times 100==\frac{i_{\mathrm{corr}1}-i_{\mathrm{corr}2}}{i_{\mathrm{corr}1}}\times 100$$
where $i_{\mathrm{corr}1}$ and $i_{\mathrm{corr}2}$ are the corrosion current densities in the presence and absence of the inhibitor, respectively.

According to the reported parameters in Table 3, higher Q-22 inhibitor concentrations decrease the anodic and cathodic corrosion current densities. In the blank acidic solution, the $i_{\mathrm{corr}}$ was the highest at all temperatures compared to the values obtained in the presence of different Q-22 concentrations. While observing the temperature effect, the $i_{\mathrm{corr}}$ increased at all tested inhibitor concentrations when the reaction temperature was raised. For instance, in the presence of 0.55 mmol·L^−^^1^ Q-22, the $i_{\mathrm{corr}}$ recorded at 20 and 70 °C were 7.28 and 252 mA·cm^−^^2^, respectively. The enormous increase in the current density (around 35 times) could be ascribed to the accelerated electrochemical reactions and metal dissolution at higher temperatures [53].

The acquired results identify Q-22 as a mixed-type corrosion inhibitor since the anodic and cathodic curves shifted towards more positive and negative potentials, respectively [54]. However, the shift in the anodic and cathodic curves at higher inhibitor concentrations was not very apparent, as in Figure 5. It is stated that inhibitors are only classified as cathodic or anodic when there is an 85 mV potential shift between the blank and inhibited solutions; otherwise, they are mixed-type inhibitors [55,56]. Therefore, the formation of a protective layer on the C-steel surface is suggested, thereby reinforcing the polarization resistance, increasing the corrosion resistance, and reducing the corrosion rates [57,58]. Indeed, the corrosion rates are reduced at higher Q-22 concentrations, elucidating a decreased affinity of C-steel with chloride ion adsorption. This reduction is attributed to the excessive accumulation of inhibitor molecules, and, hence, a boosted electron density on the surface of the C-steel. In contrast, it is observed that the corrosion rate increases at higher temperatures, which could probably be assigned to the desorption of the inhibitor molecules from the C-steel surface [59]. The minimum estimated corrosion rate was determined to be 892 mpy (mils per year) in the presence of 2.22 mmol·L^−^^1^ Q-22 inhibitor at 20 °C.

Regarding the inhibition efficiency and surface coverage, higher Q-22 inhibitor concentrations result in a higher surface coverage due to the accumulation of inhibitor molecules. Consequently, the inhibition performance is considerably improved at the highest concentration of 2.22 mmol·L^−^^1^ at all temperatures. As a result of the corrosion rate increase at higher temperatures, the efficiency was also reduced. With an increase in inhibitor concentration, the efficiency ranges from 13% to 53%, 10% to 51%, 6% to 37%, and 4% to 34% at 20, 30, 50, and 70 °C.

### 3.3. Adsorption Studies and Thermodynamic Isotherms

Adsorption isotherms assist in the study of the quasi-equilibrium adsorption of inhibitor molecules and understanding their interaction with the C-steel surface. The equilibrium constants and other thermodynamic parameters were obtained from the data obtained from the PDP measurements. Among the fitted isotherms, the Freundlich isotherm exhibited the best fitting for the surface coverage of C-steel by Q-22 inhibitor as the R^2^ values at all temperatures closely approached the unity. Figure 6 was built based on the following Equation (12).
(12)$$\log\left(Ɵ\right)=\log\left(K_{\mathrm{ads}}\right)+2.303\;n\;\log\left(C\right)$$
where Ɵ is the surface coverage, $K_{\mathrm{ads}}$ is the equilibrium constant, n is a function representing the strength of the adsorption process, and C is the concentration of inhibitor.

The equilibrium constants were calculated from the intercept of the graphs; then, these values were used to estimate the Gibbs free energy ($∆G{{}^{\circ}}_{\mathrm{ads}}$) according to:(13)$$∆G{{}^{\circ}}_{\mathrm{ads}}=-R\;T\ln\left(55.5{\;K}_{\mathrm{ads}}\right)$$
where R is the universal gas constant in J·mol^−^^1^·K^−^^1^, T is the temperature in K, and 55.5 is the water concentration in mol·L^−^^1^ [60]. Additionally, $∆G{{}^{\circ}}_{\mathrm{ads}}$ is related to the standard enthalpy ($∆H{{}^{\circ}}_{\mathrm{ads}}$) and entropy ($∆S{{}^{\circ}}_{\mathrm{ads}}$) of adsorption according to the below expression:(14)$$∆G{{}^{\circ}}_{\mathrm{ads}}=∆H{{}^{\circ}}_{\mathrm{ads}}-T\;∆S{{}^{\circ}}_{\mathrm{ads}}$$

By rearranging and compiling Equations (13) and (14), the following Van’t Hoff equation could be used to calculate the enthalpy from the slope of $\ln\left(K_{\mathrm{ads}}\right)$ versus 1/T plot (Equation (15)). After that, the entropy could also be calculated from Equation (14).
(15)$$\ln\left(K_{\mathrm{ads}}\right)=\frac{-∆H{{}^{\circ}}_{\mathrm{ads}}}{R\;T}+\frac{\;∆S{{}^{\circ}}_{\mathrm{ads}}}{R}-\ln\left(55.5\right)$$

All the thermodynamic parameters are listed in Table 4. K_ads_ is vital in providing information about the strength of interaction or the bonding between the inhibitor and metal surface [61]. It is observed that K_ads_ decreased at higher temperatures, resulting in a weak interaction and the desorption of inhibitor molecules from the C-steel surface. Additionally, the acquired $∆G{{}^{\circ}}_{\mathrm{ads}}$ values are in the range of −23 to −24 kJ·mol^−^^1^, indicating an electrostatic interaction (physisorption) between the metal and Q-22 inhibitor. Physisorption is characterized by $∆G{{}^{\circ}}_{\mathrm{ads}}\leq -20$ kJ·mol^−^^1^ [62], whereas chemisorption is considered for the condition of $∆G{{}^{\circ}}_{\mathrm{ads}}\geq -40$ kJ·mol^−^^1^. In chemisorption, there is a transfer of electrons or charge sharing between the metal and inhibitor [62]. Mixed type adsorption is classified when the $∆G{{}^{\circ}}_{\mathrm{ads}}$ values are in between the abovementioned ranges. Moreover, the standard enthalpy values are negative, confirming the exothermic nature of the adsorption process due to the entropy increase ($∆S{{}^{\circ}}_{\mathrm{ads}}$ values are positive). The entropy increase during the adsorption can be ascribed to the potential of one inhibitor molecule to substitute several water molecules, thereby increasing solvent energy [63,64].

**Table 4 molecules-27-06414-t004:** Thermodynamic parameters calculated from Freundlich isotherm fitting, as shown in Figure 6 and Figure 7.

Temperature (K)	$K_{ads}\;$	$∆G{{}^{\circ}}_{ads}\;(\mathrm{kJ}\cdot {{\mathrm{mol}}^{-}}^{1})$	$∆H{{}^{\circ}}_{ads}\;(\mathrm{kJ}\cdot {{\mathrm{mol}}^{-}}^{1})$	$∆S{{}^{\circ}}_{ads}\;(J\cdot {{\mathrm{mol}}^{-}}^{1}\cdot {K^{-}}^{1})$
293.15	256.09	−23.31	−16.40	23.55
303.15	199.99	−23.48	−16.40	23.34
323.15	128.85	−23.84	−16.40	23.04
343.15	97.03	−24.51	−16.40	23.64

### 3.4. Corrosion Kinetics Studies

The performance of the corrosion inhibitor is directly affected by its activation energy ($E_{a}$), hence estimating this parameter is of essential significance. The activation energy is influenced by the rate of anodic or cathodic reactions at various temperatures. The activation energy was calculated in the presence and absence of different Q-22 concentrations at 20, 30, 50, and 70 °C using the Arrhenius equation (Equation (16)).
(16)$$\log\;(i_{\mathrm{corr}})=\log\left(A\right)-\frac{E_{a}}{R\;T}$$
where ${\;i}_{\mathrm{corr}}$ is the corrosion current density, A is the Arrhenius constant, $E_{a}$ is the activation energy, R is the universal gas constant in J·mol^−^^1^·K^−^^1^, and T is the temperature in K.

The activation energy parameter was obtained from the slope of $\log\;(i_{\mathrm{corr}})$ versus 1/T plotted for different concentrations of Q-22 inhibitor, resulting in a straight line (Figure 8). The tabulated values of $E_{a}$ in Table 5 shows an increasing trend with the introduction of Q-22 inhibitor to the 5 M HCl solution. The higher activation energy values in the solutions containing the inhibitor suggest the physisorption of the Q-22 molecules onto the C-steel surface, causing a rise in the energy barrier of the corrosion process [65,66].

In order to determine the corrosion process entropy ($∆S^{*}$) and enthalpy ($∆H^{*}$) of activation, the transition-state equation, an alternative form of the Arrhenius equation, was followed [67].
(17)$$\mathrm{CR}=\frac{R\;T}{N\;h}e^{\frac{∆S^{*}}{R}}e^{\frac{-∆H^{*}}{R\;T}}$$
where CR is the corrosion rate, N is the Avogadro number, h is the Planck constant, $E_{a}$ is the activation energy, $∆H^{*}$ is the corrosion enthalpy, $∆S^{*}$ is the corrosion entropy, R is the universal gas constant in J·mol^−^^1^·K^−^^1^, and T is the temperature in K.

A plotting log (i/T) versus 1/T at each concentration results in straight lines, as in Figure 9. The $∆H^{*}$ and $∆S^{*}$ were calculated from the slope and intercept, respectively. The activation enthalpy increased with the inhibitor concentration, indicating a decrease in the corrosion rate depending on the activation kinetic parameters [68]. Indeed, the obtained positive values of $∆H^{*}$ reflect the endothermic nature of C-steel dissolution. Furthermore, the increase in the $∆S^{*}$ and $E_{a}$ suggests the rise in randomness as the reactants move to the activated complex [69,70]. Generally, increased $E_{a}$ and $∆S^{*}$ values at higher inhibitor concentrations confirm the physisorption nature of Q-22 adsorption onto the C-steel surface [45].

### 3.5. Surface Characterization

#### 3.5.1. Microscope Analysis

Figure 10 illustrates the surface morphology of polished C-steel coupons immersed in 5 M HCl solution for 4 h at 20 °C, in the presence and absence of the highest Q-22 concentration of 2.22 mmol·L^−1^. The micrographs show no significant defects in the polished C-steel coupons compared to the other coupons, except for the polishing scratches (Figure 10a). However, the C-steel surface (Figure 10b) exhibits severe corrosion due to the immersion in the highly corrosive 5 M HCl solution. On the other hand, 2.22 mmol·L^−1^ Q-22 inhibitor demonstrated an effective performance which can be ascribed to the significantly remarked decrease in the surface non-homogeneity and corrosion on the C-steel surface.

#### 3.5.2. AFM Analysis

The surface roughness and topography of the C-steel coupons were explored by using the 3D AFM characterization technique at the nanoscale level, as in Figure 11. AFM is a robust method for determining the effectiveness of the corrosion inhibitor through a quantitative estimation of the surface roughness. The mountain-like peaks in Figure 11b,c illustrate the surface degradation caused by the aggressive attack of ions on the exposed metal surface. The roughness of the metal surface (RMS) for the C-steel coupons immersed in the 5 M HCl medium for 4 h remarkably increased from 5.25 nm to 66.34 nm for the polished coupon (bare metal) and the coupon immersed in uninhibited HCl solution, respectively. Conversely, the C-steel surface roughness decreased by around 10.5% from 66.34 nm to 59.40 nm when 2.22 mmol·L^−1^ Q-22 was added to the 5 M HCl solution. The performed analysis proved the capability of Q-22 to be being adsorped into the C-steel surface, thus inhibiting the corrosion.

### 3.6. Eco-Toxicity Assessment

The web tools of the ADMETSAR program were exploited to assess the eco-toxic properties of Q-22 inhibitors. The properties describing the interactions between the inhibitor and both the environment and humankind, such as carcinogenicity, biodegradability, and aquatic toxicity, are shown in Table 6. These properties were selected relying on their significance to the environment and organisms and their potential impact on the inhibitor during disposal or leakage.

The results reveal that the Q-22 inhibitor molecule is safe in terms of all investigated aspects, as indicated by the probability proportions (Table 6). However, it has a 62% probability of being slightly toxic for the acute oral toxicity property. Indeed, acute oral toxicity might not be considered a dominant property, supposing the unlikeliness of oral exposure. Q-22 exhibits a 58% probability of undergoing biodegradation, hence it being classified as an environment-friendly inhibitor.

### 3.7. Quantum Chemical Calculations

Figure 12 illustrates the optimized molecular structures of Q-22, an electrostatic potential (ESP) map, the highest occupied molecular orbitals (HOMO), and the lowest unoccupied molecular orbitals (LUMO). The electron density variation among the molecular structure is visualized through colored ESP maps. Sequentially, the electrostatic potential has an ascending order indicated by red, orange, yellow, green, and blue parts [71]. The generated ESP map elucidates negative electrostatic potential (orange to yellow) over the oxygen atoms and strong positive potential (blue) over the hydrocarbon chain with the nitrogen atoms. HOMO and LUMO images display the molecular parts of the structure possessing electron-donating and accepting abilities, respectively.

The Mulliken charge distribution of Q-22 molecules is shown in Figure 13. Highest electron densities are observed over the oxygen atoms (−0.471 au). This proves that these atoms serve as active sites, boosting the interaction with and adsorption onto the metal.

The quantum chemical parameters listed in Table 7 are derived from the equations explained earlier according to DFT calculations. The energy gap (∆$E_{\mathrm{Gap}}$) is an essential parameter, indicating the reactivity of the inhibitor molecule with the metal surface. A high energy gap depicts low reactivity and interaction and, thus, a lower inhibition performance [72]. In comparison with commercial pyrimidine derivatives (PPDs) exhibiting promising inhibition efficiency for metallic surfaces, Q-22 has an ∆$E_{\mathrm{Gap}}$ which is 88% higher than the most efficient PPD structure studied by Ansari et al. [73]. Moreover, Q-22 exhibits better inhibition features than the environment-friendly AEO7 surfactant investigated by Sliem et al. [45] with an ∆$E_{\mathrm{Gap}}$ of 8.21 (Table 7). In contrast, Q-22, with an ∆$E_{\mathrm{Gap}}$ of 5.13 eV, exhibits a lower inhibition performance compared to QBBD with an ∆$E_{\mathrm{Gap}}$ of 2.05 eV. Furthermore, the slightly higher hardness (ɳ) suggests the resistance of Q-22 molecules to electron transfer. It can be noticed that the resistance of Q-22 to electron transfer is middling (=2.57 eV), with it coming between the QBBD (=2.05 eV) and AEO7 (=4.10 eV) surfactants. On the other hand, a low electronegativity (X) points out a suitable inhibitor. This applies to the case of Q-22, with it having a low value for electronegativity (X = 3.01 eV) compared to QBBD (X = 7.12 eV). The electrophilicity index $\left(\omega \right)$ predicts the energy of molecular stability after attracting electrons. The low ω value of Q-22 means it is a strong nucleophile, a vital feature of a suitable inhibitor [74]. Lastly, the high TNC corresponds to a higher donation of electrons to the unoccupied molecular orbitals of an acceptor, thereby resulting in stronger adsorption onto the metal surface [40,74].

## 4. Conclusions

The injection of corrosion inhibitors is among the most promising approaches to protecting metallic surfaces in oil and gas wells. This research study investigated the performance of the environment-friendly surfactant Q-22 as a C-steel corrosion inhibitor in a 5 M HCl medium using experimental and theoretical approaches. Initially, gravimetric analysis showed an increased efficiency at higher concentrations, i.e., a maximum IE% of 56% was achieved at 2.22 mmol·L^−1^ Q-22. Additionally, EIS and PDP measurements demonstrated effective inhibition performance at higher Q-22 concentrations, owing to the formation of a dense protective layer. Electrochemical reactions, conducted between 20–70 °C, elucidated accelerated corrosion rates and metal dissolution at elevated temperatures due to the desorption of inhibitor molecules. The minimal corrosion rate (892 mpy) was reached at the highest concentration (2.22 mmol·L^−1^) and lowest temperature (20 °C). The acquired results identified Q-22 as a mixed-type inhibitor relying on the anodic and cathodic curves. Furthermore, thermodynamic analysis showed the best fitting with the Freundlich isotherm for the surface coverage of C-steel by Q-22 molecules. All the thermodynamic parameters (K_ads_, $\;∆S{{}^{\circ}}_{\mathrm{ads}}$, $\;∆H{{}^{\circ}}_{\mathrm{ads}}$, $\;∆G{{}^{\circ}}_{\mathrm{ads}}$) were calculated, and a physisorption interaction between the metal and Q-22 inhibitor was proved. The determination of the activation energy through the study of corrosion kinetics also validated the physisorption nature of Q-22 adsorption onto the C-steel surface. SEM and AFM analyses revealed a significantly decreased surface roughness when a C-steel coupon was immersed in the inhibited HCl solution. A predicted eco-toxicity confirmed the environmentally-friendly properties of Q-22 inhibitor. Lastly, quantum calculations demonstrated some vital characteristics possessed by the Q-22 molecule for efficient inhibition performance.

## Figures and Tables

**Figure 1 molecules-27-06414-f001:**
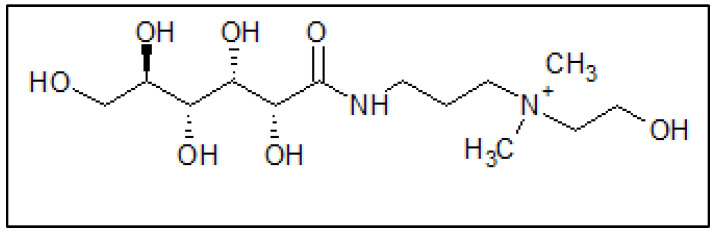
Quaternium-22 molecular structure.

**Figure 2 molecules-27-06414-f002:**
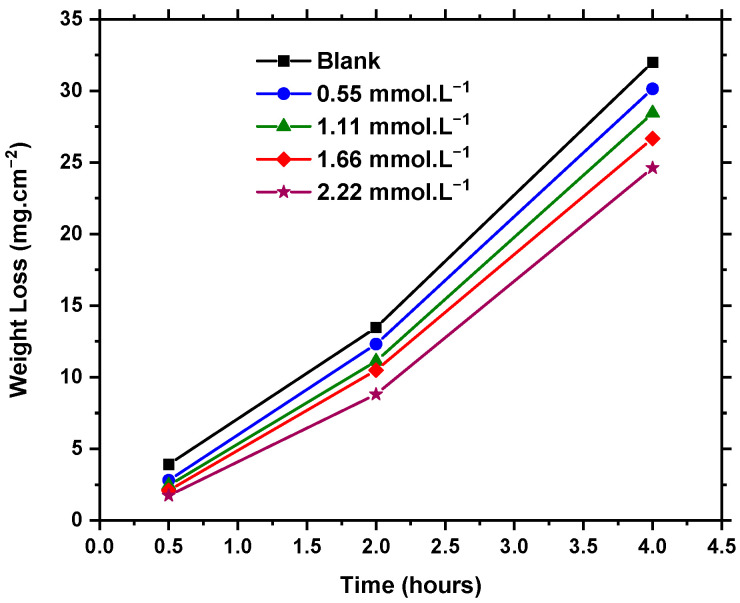
Weight loss of C-steel specimens immersed in 5 M HCl solution in the presence and absence of 0.55, 1.11, 1.66, and 2.22 mmol·L^−1^ Q-22 inhibitor at 25 °C.

**Figure 3 molecules-27-06414-f003:**
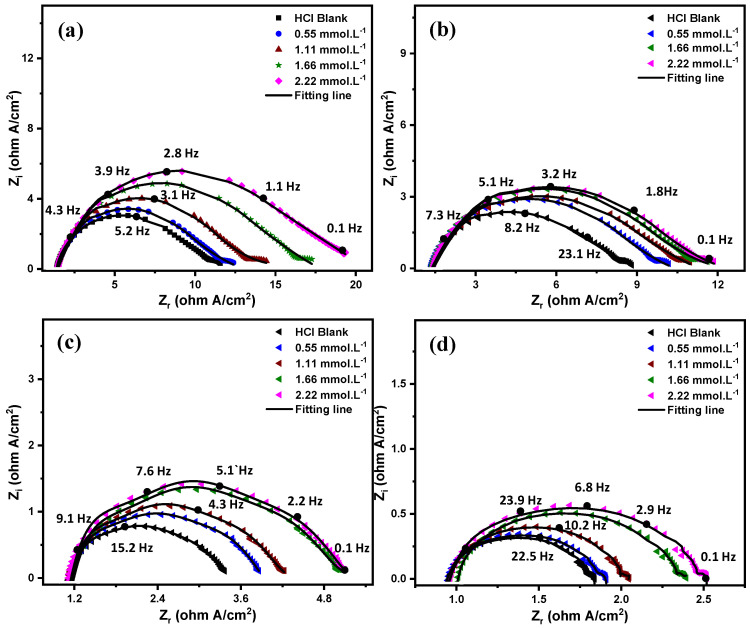
EIS Nyquist plots for C-steel in 5 M HCl in the presence and absence of 0.55, 1.11, 1.66 and 2.22 mmol·L^−1^ of Q-22 inhibitor at (**a**) 20 °C, (**b**) 30 °C, (**c**) 50 °C, and (**d**) 70 °C.

**Figure 4 molecules-27-06414-f004:**
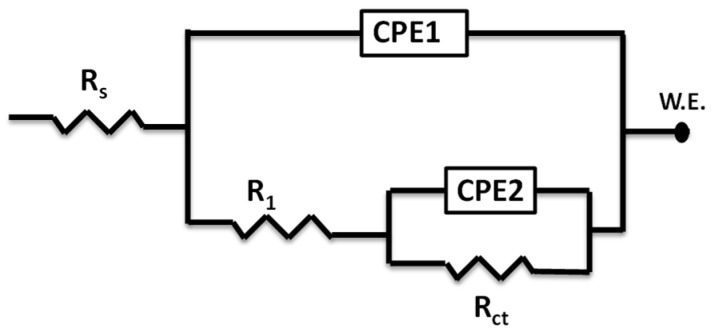
An equivalent electric circuit for EIS measured data analysis.

**Figure 5 molecules-27-06414-f005:**
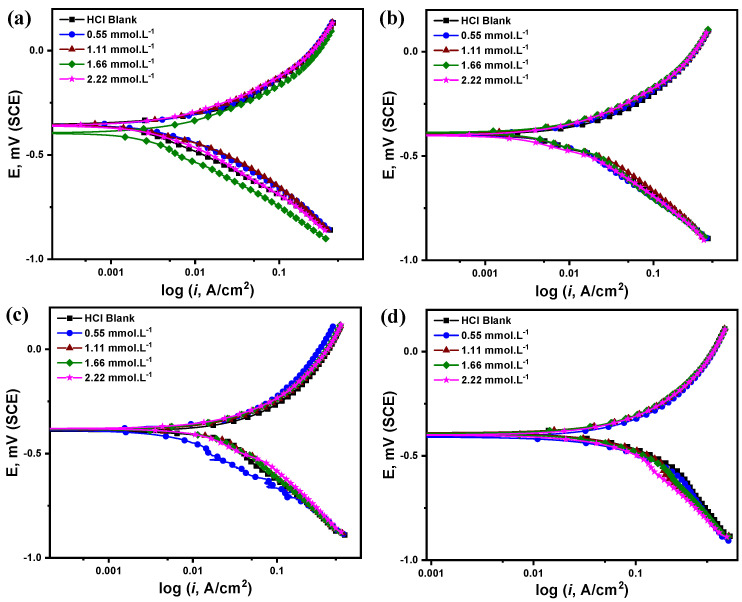
Potentiodynamic polarization curves for C-steel in 5 M HCl in the presence and absence of 0.55, 1.11, 1.66 and 2.22 mmol·L^−1^ of Q-22 inhibitor at (**a**) 20 °C, (**b**) 30 °C, (**c**) 50 °C, and (**d**) 70 °C.

**Figure 6 molecules-27-06414-f006:**
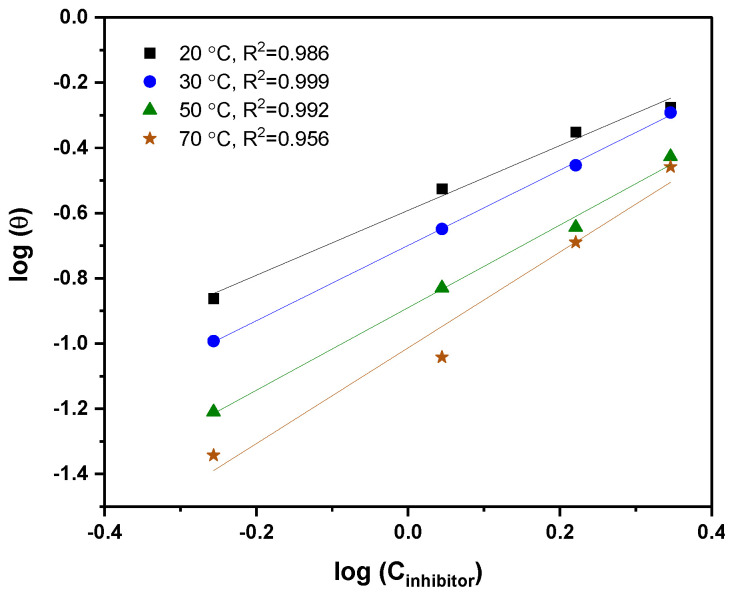
Freundlich isotherm fitting for the adsorption of Q-22 in 5 M HCl onto C-steel at 20, 30, 50, and 70 °C.

**Figure 7 molecules-27-06414-f007:**
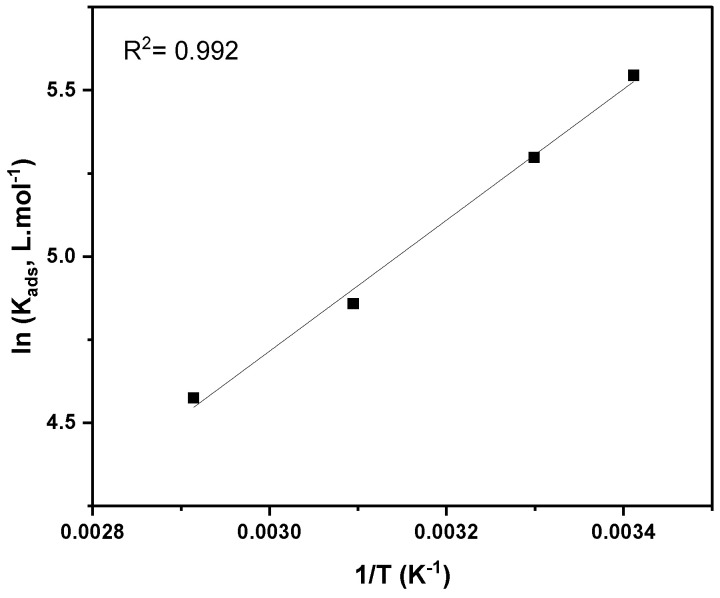
Van’t Hoff plot for adsorption of Q-22 inhibitor in 5 M HCl onto C-steel at 20, 30, 50, and 70 °C.

**Figure 8 molecules-27-06414-f008:**
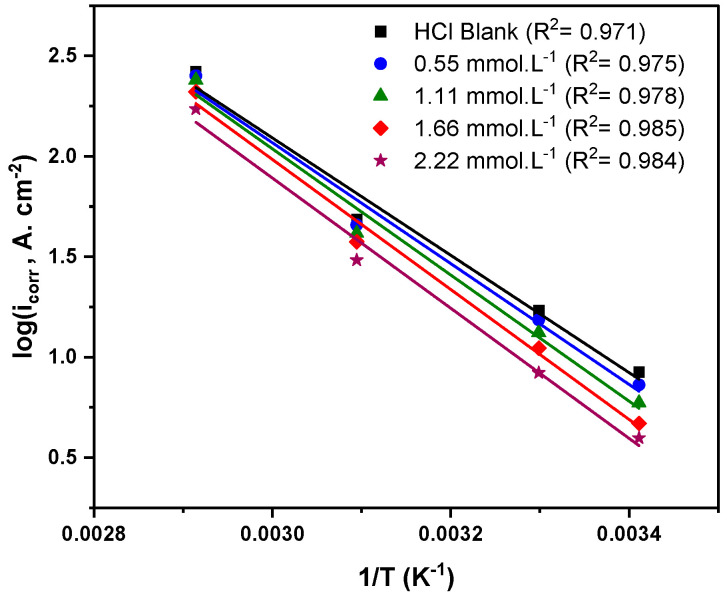
Arrhenius plots for the corrosion current densities (log i) versus 1/T for C-steel at different concentrations of the Q-22 inhibitor in 5 M HCl.

**Figure 9 molecules-27-06414-f009:**
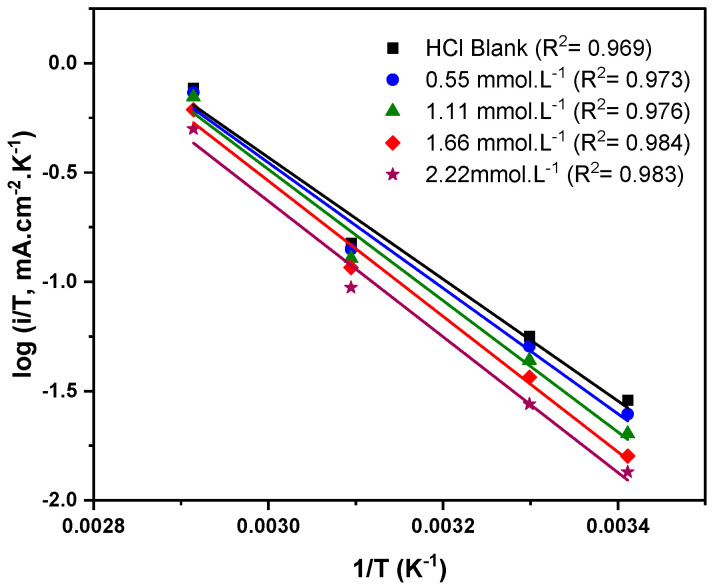
Transition-state plots of log (i/T) versus (1/T) for C-steel at different concentrations of the Q-22 inhibitor in 5 M HCl.

**Figure 10 molecules-27-06414-f010:**
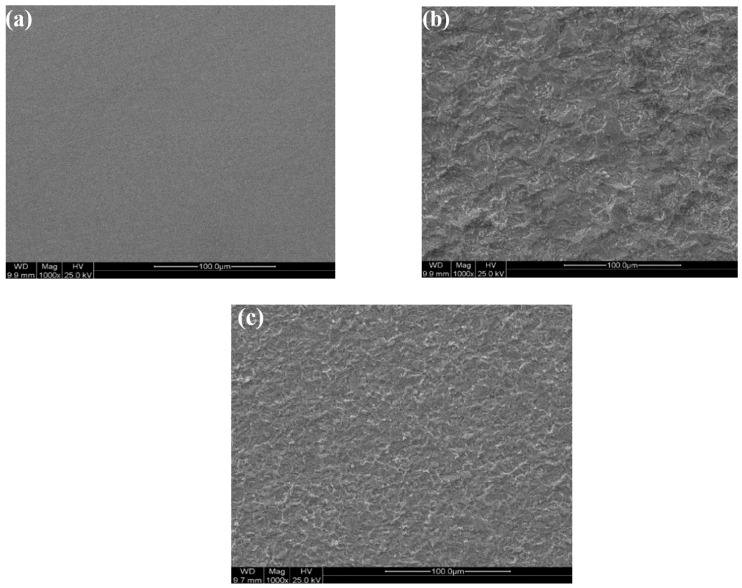
SEM micrographs of (**a**) a polished C-steel coupon in the (**b**) absence and (**c**) presence of 2.22 mmol·L^−1^ Q-22 inhibitor.

**Figure 11 molecules-27-06414-f011:**
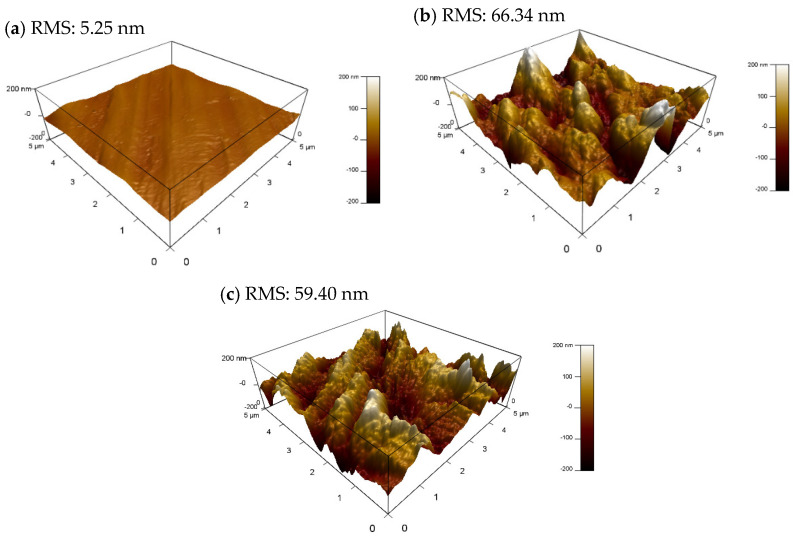
AFM images of (**a**) a polished C-steel coupon in the (**b**) absence and (**c**) presence of 2.22 mmol·L^−1^ Q-22 inhibitor.

**Figure 12 molecules-27-06414-f012:**
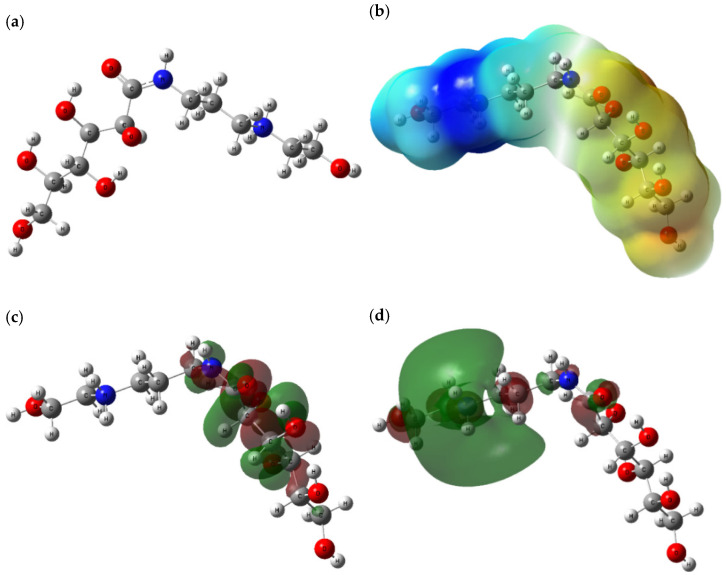
(**a**) Optimized structure, (**b**) ESP, (**c**) HOMO, and (**d**) LUMO of Q-22 inhibitor at the B3LYP/6-311+g(d,p) level of theory.

**Figure 13 molecules-27-06414-f013:**
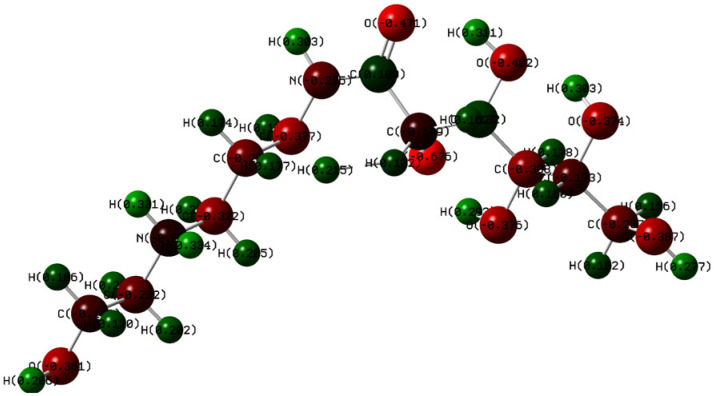
Mulliken charges of Q-22 inhibitor at the B3LYP/6-311+g(d,p) level of theory.

**Table 1 molecules-27-06414-t001:** The surface coverage and corrosion inhibition efficiency of different Q-22 inhibitor concentrations for C-steel in 5 M HCl at 25 °C.

	Surface Coverage 8			Inhibition Efficiency (IE%)		
Concentration	0.5 h	2 h	4 h	0.5 h	2 h	4 h
HCl Blank	__	__	__	__	__	__
0.55 mmol·L^−1^	0.28	0.09	0.06	28	9	6
1.11 mmol·L^−1^	0.38	0.17	0.11	38	17	11
1.66 mmol·L^−1^	0.47	0.22	0.17	47	22	17
2.22 mmol·L^−1^	0.56	0.35	0.23	56	35	23

**Table 2 molecules-27-06414-t002:** EIS parameters and corrosion inhibition efficiencies of impedance spectra of C-steel in different concentrations of Q-22 at 20, 30, 50, and 70 °C according to the equivalent electric circuit fitting.

Temperature (°C)	Conc. (mmol·L^−1^)	R_S_ (Ω·cm^2^)	R_1_ (Ω·cm^2^)	Y_1_ (µs^n^·Ω^−1^·cm^−2^)	n_1_	R_ct_ (Ω·cm^2^)	Y_2_ (µs^n^·Ω^−1^·cm^−2^)	n_2_	$C_{dl}$ (µF)	8	IE%
**20**	HCl Blank	0.67	1.50	2812	0.93	10.61	1102	0.83	443.0468471	-	-
	0.55	0.70	1.47	2832	0.64	11.95	1090	0.82	420.34	0.112	11.21
	1.11	0.70	1.41	2854	0.87	14.75	928	0.83	385.33	0.280	28.06
	1.66	0.71	1.40	2892	0.58	17.14	963	0.82	391.17	0.380	38.09
	2.22	0.70	1.37	2903	0.73	19.34	847.2	0.79	284.01	0.451	45.13
**30**	HCl Blank	0.67	1.45	2826	0.67	8.51	1247	0.83	491.48	-	-
	0.55	0.66	1.43	2846	0.48	10.21	1091	0.84	463.22	0.166	16.65
	1.11	0.69	1.37	2932	0.83	10.91	1072	0.84	459.40	0.219	21.99
	1.66	0.68	1.32	2975	0.46	11.32	1059	0.84	455.97	0.248	24.82
	2.22	0.70	1.21	3093	0.81	11.97	1002	0.84	431.47	0.289	28.90
**50**	HCl Blank	0.56	1.23	2990	0.97	3.39	1942	0.83	694.10	-	-
	0.55	0.57	1.17	3120	0.67	4.04	1676	0.84	647.24	0.160	16.08
	1.11	0.56	1.13	3170	0.96	4.23	1628	0.83	587.21	0.198	19.85
	1.66	0.57	1.09	3211	0.55	4.45	1586	0.83	575.00	0.238	23.82
	2.22	0.56	1.06	3254	0.83	4.66	1585	0.81	501.21	0.272	27.25
**70**	HCl Blank	0.48	0.97	3260	0.81	1.83	2467	0.95	1856.6	-	-
	0.55	0.50	0.98	3242	0.75	1.91	2385	0.94	1690.5	0.041	4.18
	1.11	0.49	0.95	3580	0.98	2.04	2261	0.93	1508.2	0.102	10.29
	1.66	0.49	1.01	3272	0.53	2.28	2254	0.92	1425.2	0.197	19.73
	2.22	0.49	0.93	3690	0.87	2.41	2201	0.91	1310.9	0.240	24.06

**Table 3 molecules-27-06414-t003:** Potentiodynamic polarization parameters and corrosion inhibition efficiencies of C-steel in different concentrations of Q-22 at 20, 30, 50, and 70 °C according to Tafel fit.

Temperature (°C)	Conc. (mmol·L^−1^)	−E_corr_ (mV, SCE)	i_corr_ (mA·cm^−2^)	$\beta_{a}$ (nV/decade)	$\beta_{c}$ (nV/decade)	$R_{p}$ $\left(Ω\cdot {\mathrm{cm}}^{2}\right)$	CR (mpy)	8	IE%
**20**	HCl Blank	354	8.44	230.50	288.40	6.59	1583	-	-
	0.55	357	7.28	212.50	275.90	7.16	1632	0.14	14
	1.11	357	5.92	193.30	248.50	7.97	1163	0.30	30
	1.66	360	4.68	185.20	254.80	9.95	1195	0.45	45
	2.22	360	3.96	177.20	250.30	11.38	892	0.53	53
**30**	HCl Blank	399	17.10	235.30	321.00	3.45	3160	-	-
	0.55	396	15.36	235.00	328.10	3.87	3075	0.10	10
	1.11	392	13.26	230.70	288.10	4.20	2715	0.22	22
	1.66	388	11.08	228.60	316.90	5.20	2598	0.35	35
	2.22	401	8.36	222.10	274.60	6.38	1813	0.51	51
**50**	HCl Blank	393	48.60	382.70	579.50	2.06	37130	-	-
	0.55	388	45.60	366.30	520.70	2.05	25240	0.06	6
	1.11	385	41.40	345.70	499.20	2.14	17760	0.15	15
	1.66	384	37.56	323.00	441.30	2.16	13170	0.23	23
	2.22	380	30.42	249.40	344.80	2.07	9277	0.37	37
**70**	HCl Blank	390	264.00	992.00	1041.00	0.84	121300	-	-
	0.55	391	252.00	904.50	1038.00	0.83	93680	0.05	5
	1.11	390	240.00	830.40	1159.00	0.88	65410	0.09	9
	1.66	384	210.00	763.40	913.40	0.86	62210	0.20	20
	2.22	380	172.00	631.80	927.20	0.95	59860	0.35	35

**Table 5 molecules-27-06414-t005:** The activation energy of C-steel in 5 M HCl in the presence and absence of different Q-22 concentrations as obtained from the Arrhenius plots.

Concentration (mmol·L^−1^)	$E_{a}\;(\mathrm{kJ}\cdot \mathrm{mol}{}^{-1})$	$∆H^{*}\;(\mathrm{kJ}\cdot \mathrm{mol}{}^{-1})$	$∆S^{*}\;(J\cdot \mathrm{mol}{}^{-1}\cdot K{}^{-1})$
HCl Blank	55.94	53.31	151.67
0.55	57.62	54.98	156.22
1.11	60.14	57.51	163.22
1.66	61.97	59.34	167.70
2.22	62.05	59.41	166.15

**Table 6 molecules-27-06414-t006:** Eco-toxic properties of Q-22 inhibitor [34,35].

Carcinogenicity	Eye Irritation	Ames Mutagenesis	Acute Oral Toxicity (Class III)
0.9 (safe)	0.95 (safe)	0.67 (safe)	0.62 (slightly toxic)
**Honey bee toxicity**	**Biodegradability**	**Fish aquatic toxicity**	**Water solubility (LogS)**
0.72 (safe)	0.58 (safe)	0.79 (safe)	−1.86 (soluble)

**Table 7 molecules-27-06414-t007:** Quantum parameters of Q-22 calculated at the B3LYP/6-311+g(d,p) level of theory and compared with other green surfactants in the literature (QBBD and AE07 calculated at B3LYP/6-31+g(d,p) and B3LYP cc-pvdz basis set, respectively).

	Q-22 (This Study)	QBBD [31]	AEO7 [46]
**HOMO (eV)**	−5.57	−8.17	-
**LUMO (eV)**	−0.44	−6.12	-
$∆E_{Gap\;\left(LUMO-HOMO\right)}\;$ (eV)	5.13	2.05	8.21
**I (eV)**	5.57	-	6.89
**A (eV)**	0.44	-	−1.31
**ɳ** **(eV)**	2.57	2.05	4.10
**Χ (eV)**	3.01	7.12	2.79
**ω (eV)**	11.60	-	-
**TNC (eV)**	−5.59	-	-

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
