# Peer review of "An Eco-Friendly Quaternary Ammonium Salt as a Corrosion Inhibitor for Carbon Steel in 5 M HCl Solution: Theoretical and Experimental Investigation"

_molecules, 2022, doi:10.3390/molecules27196414_

Round 1
Reviewer 1 Report
1. Even if under the same calculation method, setting parameters are different, the calculation results of the same substances will also be different, and such difference may affect the comparison results. Therefore, it is suggested to list the calculation parameters of QBBD and AEO7 in Table 7 to improve the credibility of the paper.
2. The free chlorine atom in the q-22 molecular structure looks very special. Please make sure if it is connected to the main chain of the structure. If chlorine is supplied by a corrosive environment, it will not appear in the structural formula.
3. There is a large number of editing errors (Error! Reference source not found.)Please correct them.
Author Response
Comments and Suggestions for Authors
- Even if under the same calculation method, setting parameters are different, the calculation results of the same substances will also be different, and such difference may affect the comparison results. Therefore, it is suggested to list the calculation parameters of QBBD and AEO7 in Table 7 to improve the credibility of the paper.
Response 1: The authors appreciate the reviewer comment. The calculation method has been added to Table 7.
- The free chlorine atom in the q-22 molecular structure looks very special. Please make sure if it is connected to the main chain of the structure. If chlorine is supplied by a corrosive environment, it will not appear in the structural formula.
Response 2: The authors have revised the structure and removed the chlorine from the corrosive environment. The quantum chemical parameters and figures are updated accordingly.
- There is a large number of editing errors (Error! Reference source not found. Please correct them.
Response 3: The authors are grateful for the reviewer's comment and references were updated and corrected.
Reviewer 2 Report
Dear Authors,
I have started to read your manuscript and I have realized that you have at least 28 issues with the references (see "Error!". So, unfortunately I cannot review your manuscript.
Major issues:
Please check Eq. (2) as A might be in inches. Also provide a reference for this equation to avoid any uncertainty.
Section 3:There is an error with the references.
Minor issues:
Line 18: “The maximum estimated inhibition efficiency was 53%” this refers to electrochemical measurements? Please clarify this in the text.
lines 57-59: This composition, aromatic groups and pi-electrons also refer to conducting polymers that are known corrosion inhibitors. So you can quote them here?
Line 90: Please be more precise here: “a high efficiency of 98.9%...”
Line 117: There is an apparent error with the reference to Figure.
Line 122: Insert % for iron to facilitate reading process.
Line: 157; Who many points per decade?
Line 164: AFM in which mode? Please clarify.
Author Response
Comments and Suggestions for Authors
Dear Authors,
I have started to read your manuscript and I have realized that you have at least 28 issues with the references (see "Error!". So, unfortunately I cannot review your manuscript.
The authors do thank the reviewer for his/her precious comments which enriched the manuscript. We have done all the corrections listed below and highlighted that in the manuscript.
Major issues:
Please check Eq. (2) as A might be in inches. Also provide a reference for this equation to avoid any uncertainty.
Response 1: The authors agree with the reviewer comment as corrosion rate was measured by Mils penetration per year (MPY) which equal to one thousandth of an inch.
Section 3: There is an error with the references.
Response 2: The authors be grateful for the reviewer comment and all errors have been corrected.
Minor issues:
Line 18: “The maximum estimated inhibition efficiency was 53%” this refers to electrochemical measurements? Please clarify this in the text.
Response 1: The authors have been rewritten this paragraph and highlighted it with green in the abstract
lines 57-59: This composition, aromatic groups and pi-electrons also refer to conducting polymers that are known corrosion inhibitors. So you can quote them here?
Response 2: The authors have mentioned conducting polymers in the manuscript.
Line 90: Please be more precise here: “a high efficiency of 98.9%...”
Response 3: The authors have been modified the sentences.
Line 117: There is an apparent error with the reference to Figure.
Response 4: The authors corrected the errors in the figures number
Line 122: Insert % for iron to facilitate reading process.
Response 5: Done
Line: 157; Who many points per decade?
Response 6: The authors counted 10 point per decades for the EIS measurements
Line 164: AFM in which mode? Please clarify.
Response 7: The authors has utilized the non-contact mode of AFM measurements. Hence, the tip vibrates close to the surface and experiences attractive forces, so the amplitude of vibration and vibration frequency are lowered.
Round 2
Reviewer 2 Report
Dear Authors, the manuscript is now more readable as there are fewer issues with figures references. The corrosion inhibitors and their applications are interesting topics due to their constant application in the industry thus, this topic is interesting for the Journal. However, the paper is very expensive at some point, and thus; the reader might lose focus. Therefore, it should be shortened.
The main issue in the paper is the interpretation of EIS data. Specifically, you did not mark the frequency values. Also, you comment on capacitive behavior, but the values in Fig.3 are positive!? Thus, the proposed EEC and the presented fitted data do not make sense at some points. If you made a mistake by not writing negative signs before Zi, then you have inductive behavior at the high frequencies? But, the applied EEC does not possess inductive properties. Furthermore, in some subfigures, the presence of 2 semicircles can be observed but the applied EEC can produce only one semicircle. So, it appears that the EIS study is in principle doubtful.
Furthermore, DFT debate (see lines 482-490) should be rephrased and simplified. So, please clearly explain the impact of band gap on 1) metal-Q-22 interactions, 2) inhibition efficiency and 3) clearly explain the advantages of Q-22 over QBBD and AEO7 using numbers presented in Table 7.
Overall, this is an extensive manuscript that offers a fair amount of data. However, there are several drawbacks that need to be resolved. Specifically, the EIS study is rather doubtful and should be rewritten.
Minor issues:
In Abstract you use Q22 while in the rest of the text you write Q-22.
Line 205: Again: Error! “Reference source not find”
Line 94: Check “+” sign. It should not be in subscript.
Line 98: Please check “TritonX100” for typos. It should be “Triton X-100”?
Line 137: “%” is missing at the right side?
Line 157: “taking 10 points per decade”?
Line 162: So, you assume that there is only 5 % of error?
Line 270: “%” is missing at the right side?
Major issue:
Line 282: “Looking at the results, the optimum concentration of Q-22 is 2.22 mmol/L…. “. This is not correct as this is the highest applied concentration. So, this is not optimal.
In Table 3, what is the meaning of “Ba” and “Bc”? Typo?
